# Shrimp Larvae Counting Based on Improved YOLOv5 Model with Regional Segmentation

**DOI:** 10.3390/s24196328

**Published:** 2024-09-30

**Authors:** Hongchao Duan, Jun Wang, Yuan Zhang, Xiangyu Wu, Tao Peng, Xuhao Liu, Delong Deng

**Affiliations:** Centre for Optical and Electromagnetic Research, South China Academy of Advanced Optoelectronics, South China Normal University, Guangzhou 510006, China; 2021024069@m.scnu.edu.cn (H.D.); 2020023787@m.scnu.edu.cn (J.W.); 2022024121@m.scnu.edu.cn (X.W.); 2023024193@m.scnu.edu.cn (T.P.); 2023024174@m.scnu.edu.cn (D.D.)

**Keywords:** shrimp larvae counting, YOLOv5, regional segmentation, attention mechanism, repeat shrimp removal

## Abstract

Counting shrimp larvae is an essential part of shrimp farming. Due to their tiny size and high density, this task is exceedingly difficult. Thus, we introduce an algorithm for counting densely packed shrimp larvae utilizing an enhanced You Only Look Once version 5 (YOLOv5) model through a regional segmentation approach. First, the C2f and convolutional block attention modules are used to improve the capabilities of YOLOv5 in recognizing the small shrimp. Moreover, employing a regional segmentation technique can decrease the receptive field area, thereby enhancing the shrimp counter’s detection performance. Finally, a strategy for stitching and deduplication is implemented to tackle the problem of double counting across various segments. The findings from the experiments indicate that the suggested algorithm surpasses several other shrimp counting techniques in terms of accuracy. Notably, for high-density shrimp larvae in large quantities, this algorithm attained an accuracy exceeding 98%.

## 1. Introduction

Shrimp farming has become a vital economic sector within aquaculture, contributing significantly to the growth of the fish industry [1]. Counting shrimp larvae is an essential task in the shrimp farming process, as it assists farmers in determining the reproductive rate and accurately estimating the production potential [2]. In addition, it helps evaluate fertility, control the density of cultivation, and manage transport sales [3,4]. However, due to the tiny size, great flexibility, and dense numbers of shrimp larvae, counting them presents a significant challenge [5]. Currently, the process is predominantly carried out by hand, making it a time-consuming, labor-intensive, and somewhat imprecise task [6]. Thus, developing an innovative approach to overcome the current difficulties in counting shrimp larvae is of the utmost importance.

The progress in the counting of shrimp larvae consists primarily of manual counting [7], photoelectric detector counting [8,9,10,11], counting based on conventional image processing [12], and deep learning-based counting [13]. The first technique is highly dependent on the operator’s expertise, resulting in substantial differences in the accuracy and efficiency of the counting process. In addition, the process of manual counting may harm shrimp larvae. Photoelectric devices for counting shrimp larvae work by shining light on the larvae and determining their number based on the light reflected back to the sensors [8]. Although this method significantly improves both the speed and accuracy, these counters can be affected by various environmental factors, such as lighting conditions and water quality, resulting in considerable counting errors. Furthermore, photoelectric devices are generally effective for shrimp larvae of certain sizes, which reduces their accuracy when dealing with larvae of different sizes.

Recently, computer vision technology has been applied in various fields [14], such as target detection, image super-resolution, and population counting [15]. Regarding the counting of shrimp larvae [16,17,18,19], traditional image processing techniques primarily utilize image segmentation and object detection methods to recognize and count target images. Kesvarakul et al. counted shrimp larvae in images by converting them into binary images with a threshold [20]. However, this technique is only effective when dealing with a small quantity of larvae and when their images are clear. Thai et al. used image segmentation and contour tracking methods for the counting of shrimp larvae [21]. This method is able to count individual shrimp larvae, although it struggles to accurately distinguish shrimp larvae which are stuck together. For populations with fewer shrimp larvae, counting was performed using algorithms, such as segmentation paired with the Canny edge detection method and blob processing approaches [22]. Although this technique offers an enhancement over previous algorithms, it still does not address the issue of shrimp and larvae adhering to each other. Traditional image processing-based counting techniques require prior knowledge to manually adjust image features, leading to poor accuracy in scenarios with complex backgrounds. Furthermore, the lack of generalization in these models hinders the accurate detection and counting of shrimp larvae in different scenarios.

Subsequently, deep learning-based [23,24,25,26] counting methods use trained object recognition models to identify targets within images. Armalivia et al. used the You Only Look Once version 3 (YOLOv3) algorithm to count shrimp larvae [27], achieving an average counting accuracy of 96% in low-density populations of around 100 larvae (in a circular area with a diameter of 40 cm), but a larger number of larvae could not be identified. Hu et al. used deep learning and a density map of shrimp larvae to estimate the populations of approximately 1000 larvae (in a 24 cm × 33 cm area) [13]. However, this approach does not accurately identify each individual larva. Hence, devising an automated and accurate method for counting shrimp larvae in densely populated areas remains a difficult challenge. Object detection algorithms based on deep learning mainly include Regions with Convolutional Neural Networks (R-CNN) and multiple versions of YOLO, among which YOLOv5 is a mature algorithm among target recognition algorithms which has the advantages of being lightweight and having a fast inference ability. However, in the detection and recognition of small targets, YOLOv5 is not effective, and thus we need to improve YOLOv5.

In this study, we investigate the enhancement of deep learning techniques to address the challenge of counting shrimp larvae in highly populated environments. In particular, we propose an algorithm which uses regional segmentation along with an improved YOLOv5 model to accurately count shrimp larvae, especially in highly dense environments. The remainder of this paper is structured as follows. In Section 2, we give the related work to highlight our motivation. In Section 3, we describe the details of the proposed shrimp larvae counting method. Section 4 presents the experimental results and further validates the superiority of the algorithm through controlled experiments. Section 5 gives our conclusions.

The main contributions are described as follows:•We build an automatic shrimp collecting platform to obtain the corresponding dataset and provide a counting algorithm based on improved YOLOv5 with region segmentation.•The C2f and attention mechanism are embedded in the YOLOv5 model, improving the detection ability for small shrimp.•The segmentation of regions aims to boost the proportion of pixels representing shrimp larvae in a visual image.•Experimental results prove that the proposed algorithm can identify shrimp larvae with high accuracy under the difficult circumstance of overlap between large numbers of shrimp and different light intensities.

## 2. Related Work

There are many challenges in the counting of shrimp larvae. First, shrimp larvae are small in size, making them relatively difficult to detect. In addition, shrimp larvae move rapidly, and their pixel values in images differ depending on their depth in the water. Conventional threshold segmentation techniques may easily overlook shrimp larvae which are at the bottom of the water.

As shown in the Figure 1, there are three states, including isolated, clumped, and overlapping shrimp larvae. Isolated shrimp larvae remain unaffected by the presence of other larvae and can be detected through conventional image processing approaches or deep learning methods. Clumped shrimp larvae consist of several larvae attached to each other, resulting in an underestimated count when using connected component analysis. This problem can be resolved through the application of watershed algorithms and distance transformation. However, these techniques necessitate precise threshold configurations and do not generalize well. Overlapping shrimp larvae appear when larvae at varying water depths align at the same spot in the image. The aforementioned watershed algorithm is ineffective in addressing this scenario. Utilizing concave point algorithms for separation might erroneously fragment the shrimp tails, causing overcounting of the shrimp. Consequently, conventional image processing techniques prove inadequate for identifying these instances, necessitating the use of deep learning approaches to accurately discern and identify the various states of shrimp larvae.

Minimizing the dimensions of the input image can enhance the accuracy of receptive field detection [28]. Two widely used techniques for decreasing the size of an image are image scaling and cropping. Image scaling often employs downsampling [29], which may lead to data loss. Another approach is image segmentation, which can be performed either randomly or in a consistent manner. Random segmentation employs random functions to determine the coordinates for the top-left and bottom-left sections of the local image of shrimp larvae [30]. Then, the coordinates for the top-right and bottom-right sections are determined according to the dimensions of the local image. In addition, the cropping function is applied to extract the local image. Beginning in the upper-left corner of the original image, fixed segmentation continues with sequential segmentation based on the dimensions of the local image. For regions along the edges which are smaller than the segmented dimensions, the true size of the image is maintained. Random segmentation might result in repeated recognition of the same regions, thereby increasing the training burden on the model. Consequently, this research uses fixed segmentation to handle the initial images.

The estimation of density maps is also a commonly used counting method [23] which estimates the number of objects by generating a density map. This is often applied in crowd counting [31]. However, this method cannot accurately provide the geometric and positional information of objects. When the detection of small objects does not require positional information, the method can provide good counting results. In shrimp larvae counting, when the larvae density is relatively low, positional and shape information may not be critical, and both density map estimation and object detection can achieve accurate counts. However, as the larvae density increases, with overlapping larvae at different water depths, positional and shape information become important. In high-density areas, density map estimation is prone to misjudgments, leading to a decrease in count accuracy. By incorporating position and shape information, object detection methods can achieve better counting accuracy. In terms of runtime, density map estimation models are more complex and require a longer processing time, while object detection methods are faster, making them more suitable for quick shrimp larvae counting.

YOLO [32] is considered one of the best choices for object detection models due to its speed and superior accuracy compared with R-CNN [33]. YOLO pioneered an innovative approach by utilizing convolutional neural networks on a complete image to predict object classes and bounding boxes via direct feature regression. Over the years, YOLO has evolved into several versions. YOLOv1 uses a unified detection approach for object localization and classification tasks [32]. Subsequently, YOLOv2 improves on version 1 by having better accuracy, a faster speed, and the ability to recognize more objects [34]. YOLOv3 improves the object detection speed by implementing multiscale prediction [35], optimizing the core network, and refining the loss function. YOLOv4 introduces a fast and effective object detection model which substantially cuts down computational expenses, making it more compatible with general-purpose devices and those with hardware constraints. YOLOv5 brings significant advancements, including the CSPDarknet backbone and mosaic augmentation, balancing speed and accuracy [36,37,38,39,40,41,42]. YOLOv7 improves the structure of the extended efficient layer aggregation network, revises the model architecture, and introduces an efficient label assignment strategy, which increases detection performance while decreasing the number of model parameters [43]. Currently, researchers are constantly exploring the architectural design of YOLO, with the latest version being YOLOv10 [44]. It has achieved state-of-the-art performance and efficiency on various model scales. In the experimental section, we describe the performance of YOLOv10 and conduct shrimp larvae counting tests using this model. However, the YOLOv10 model is complex and has high hardware resource requirements, making its hardware deployment challenging. One of the primary objectives of this study is to deploy the algorithm on hardware to achieve an integrated system for shrimp larvae image acquisition and counting. YOLOv10 does not meet the requirements of our design. YOLOv5 stands out from its predecessors with its lighter architecture, faster inference times, and more developed ecosystem which offers enhanced compatibility [45]. Moreover, YOLOv5 has low hardware resource requirements, is relatively mature in terms of hardware deployment, and is easier to implement. Therefore, we selected YOLOv5 as the baseline for our study. However, in scenarios which involve the detection of small objects, such as shrimp larvae, the use of several convolutional layers may lead to missing small targets, resulting in reduced pixel occupancy for small objects. To address this problem, we upgraded the YOLOv5 backbone network to improve its ability to identify small targets.

## 3. Materials and Methods

For a more precise counting of shrimp larvae in situations with large numbers and a high density, we suggest an algorithm based primarily on regional segmentation and an improved YOLOv5 model. The flow chart of the algorithm can be viewed in Figure 2.

To begin with, we divide the whole original image into smaller segments using a region-based segmentation algorithm. Subsequently, we employ an enhanced YOLOv5 model to recognize shrimp larvae in each of these isolated images separately. Then, all segmented image blocks are reassembled into a complete image. A repeat count removal method is proposed to address the issue of repeat counting in the stitching position, ensuring that accurate counting of shrimp larvae can be achieved throughout the image.

### 3.1. Image Collection

The quality of the original images, which is one of the key factors for computer vision, will affect the accuracy of the counting of shrimp larvae. For the best accuracy in counting the larvae, we designed and fabricated a shrimp larvae image acquisition device as shown in Figure 3.

Shrimp larvae are placed in a semi-transparent plastic square bucket, and white LEDs with light guide plates are added around the side wall and the bottom to provide uniform illumination. Shrimp larvae images are captured using an industrial camera (MV-CE060-10UM (Hikvision, Hangzhou, China)) with an 8 mm lens under supplementary white light. The resolution of the camera is 3072 × 2048 pixels for a field of view of about 40 cm × 30 cm, and the effective area we used for the shrimp larvae was 30 cm × 30 cm.

The shrimp larvae dataset used in this experiment was collected from on-site photography at the Hongkai Shrimp Larvae Farm in Zhuhai City, Guangdong Province, China. Figure 4a shows the real shrimp larvae farming environment. The staff used a fishing net to extract shrimp larvae from the pool (as shown in Figure 4b) and then placed the collected larvae in our custom-built shrimp larvae image acquisition device as shown in Figure 4c.

### 3.2. Region Segmentation

The receptive field refers to the size of the region in the input image which corresponds to each pixel in the output feature map [46]; that is, a single point on the feature map corresponds to a specific region on the input image. The size of the area of the receptive field directly affects the detection accuracy [47,48]. The formula for calculating the receptive field is as follows:(1)$$R_{i}=\left(R_{i+1}-1\right)\times St_{i}+Ks_{i}$$
where $R_{i}$ represents the receptive field on the *i*th convolutional layer, $R_{i+1}$ is the receptive field on the i+1th layer, $St_{i}$ is the stride of the convolution, and $K_{S}$ is the size of the convolutional kernel for the current layer.

When the original image of shrimp larvae (3072×2048 pixels) is used as the input image, the corresponding receptive field size is 38×38 pixels. This indicates that each point on the feature map corresponds to a area of 38×38 pixels on the original image, which is significantly larger than the size of the shrimp larvae (approximately 10×10 pixels). As a result, information on shrimp larvae would be overlooked or covered, greatly reducing the detection accuracy.

To address this issue, it is necessary to reduce the size of the receptive field to improve the shrimp larvae detection performance. Therefore, we adopted the region segmentation method to divide the original image, reducing the size of the input image and thereby decreasing the receptive field area. For example, the original image of shrimp larvae (as shown in Figure 5a) was divided into multiple image blocks (100×100 pixels for each, as shown in Figure 5b) for subsequent labeling and training.

In the whole image, the proportion of shrimp larvae in terms of pixels in the whole image is extremely small. Therefore, we used segmentation technology to obtain multiple local regions of shrimp larvae, making feature extraction easier.

Figure 6 shows the feature maps selected from the 17th layer of the YOLOv5 model for both the full image and the local image. To facilitate comparison, we extracted the same region from the full image as that in the local image. From Figure 6, it is evident that the full image did not accurately identify local regions, while the local image allowed a more precise extraction of features from the shrimp larvae.

### 3.3. The Improved YOLOv5

The core idea of YOLOv5 is to convert the target detection problem into a regression problem, predicting the boundary boundaries and categories of the target in the image. It has high accuracy and a fast inference ability, making it one of the best-performing target detection models available today. However, the tiny size of the shrimp larvae, along with various body positions such as their overlap, leads to a decrease in the identification accuracy of the YOLOv5 model. Therefore, the recognition accuracy can be improved by increasing the classification performance of features.

As shown in Figure 7, the C2f module (Figure 8) was used to replace the C3 module. In the backbone network of the YOLOv5 algorithm, the main function of the C3 module is to extract image features and enhance the learning capacity of the convolutional network. The C3 module cannot meet the requirements for the detection of small targets such as shrimp larvae and needs further improvement to enhance the feature extraction capabilities of the model. The C2f module processes the input data using two convolutional layers, which assist in extracting features at varying levels and degrees of abstraction. This improves the feature extraction efficiency and simultaneously reduces the network weight, facilitating more abundant gradient flow information [49]. In contrast to the C3 module, the C2f module is more lightweight, has reduced computational demands, and demonstrates robust feature extraction capabilities.

To further enhance the recognition capability of the model, the convolutional block attention module (CBAM) [50] is introduced into the YOLOv5 framework. The CBAM consists of the channel attention and spatial attention modules, as shown in Figure 9. For an input *F*, the global average and maximum pooling operations are first applied to obtain global information for each channel. Subsequently, two fully connected layers are utilized to produce the channel attention vector. These layers apply weights to the input feature *F* per channel, which yields the refined feature $F_{1}$. In the spatial attention module, the feature map is combined to capture spatial information. Furthermore, the feature $F_{1}$ undergoes a convolution layer 3×3, resulting in the creation of a spatial attention map via the sigmoid function. The spatial attention map is subsequently used to enhance the feature $F_{1}$, leading to production of the fused feature $F_{2}$. The attention module improves the depiction of shrimp larvae features across different conditions.

Figure 10 displays the feature map of both the enhanced YOLOv5 network and the original YOLOv5 network. Clearly, the proposed approach provides more precise counting outcomes even in instances of densely packed shrimp larvae.

### 3.4. Repeat Count Removal via Stitching

The method described in Section 3.2 allows for segmentation of the entire image. The method described in Section 3.3 can identify shrimp larvae in each segmented image block. However, identical shrimp larvae can be segmented into neighboring image blocks and detected at the same time (in Figure 11), resulting in reduced counting accuracy. To address the issue of duplicate counts, an additional detection model was developed.

Figure 11a shows a shrimp larva divided vertically into two sections and identified in two separate subimages. Figure 11b presents two shrimp larvae segmented horizontally into two subimages and identified. Figure 11c shows shrimp larvae cut horizontally and vertically, resulting in four distinct subimages. Figure 12 shows a schematic illustration of the segmentation of shrimp larvae, helping in the evaluation of the segmentation configurations.

In Figure 12, the red boxes represent the detection boundary boxes of the detection output, and the black lines represent the cropping boundaries of the image. Taking into account the spatial relationship between the detection boundary boxes and the image cropping boundaries in the output, nine distinct scenarios can be identified: undivided shrimp larvae (N), top edge touching (U), bottom edge touching (D), right edge touching (R), left edge touching (L), right and bottom edges touching (RD), right and top edges touching (RU), left and bottom edges touching (LD), and left and top edges touching (LU). A set *S*, where S={N,D,U,R,L,RD,RU,LD,LU}, was defined to store all detection output. By traversing the detection outputs and checking the boundaries, the detection results were placed in the corresponding sets.

To avoid duplication, we examined the shrimp larva detection boxes in neighboring regions to determine whether the detected shrimp larvae were identical. If several detection boxes corresponded to each other, as illustrated in Figure 13, then $\left(x_{i},y_{i}\right)$ are the coordinates of the vertices of the detection frame, and the detection box at the bottom which intersects the cropping boundary coincides with the two detection boxes at the top, which also intersect with the cropping boundary. Therefore, an overlap ratio must be implemented to facilitate the selection process. We define a variable $R_{o}$ to represent the percentage of overlap between detection frames, whose formula is as follows:(2)$$R_{o}=\frac{x_{4}-x_{1}}{x_{2}-x_{3}}$$
where $x_{1},x_{2},x_{3},x_{4}$ are the horizontal coordinates of the detection frame.

By calculating the overlap ratio, the detection box with the highest overlap ratio is selected. After determining the matching object, a minimum bounding rectangle is used to completely enclose the two detection boxes, thus identifying the deduplication area (indicated in Figure 14a, for example). The two red detection boxes in Figure 14b correspond to a typical repeat count case (according to Figure 11b). We determined whether they were connected by examining the coordinates of the two detection boxes. Once the connected boxes were identified, we used a minimum enclosing box (indicated by the green rectangle in Figure 14a,c) to completely cover the two red detection boxes and define the deduplication area.

Once the deduplication area is identified, it is fed back into the enhanced YOLOv5 network mentioned in Section 3.3 to accurately count the shrimp larvae in this region. This process ensures that duplicate counts are removed from the entire composite image of shrimp larvae.

## 4. Experimental Results and Analysis

### 4.1. Implementation Details

In the experiment, we collected 20,000 valid local images of shrimp larvae, with 15,000 allocated for training and 5000 for testing. For annotating the dataset, the conventional labeling approach (Figure 15a) involves describing all the shrimp larvae, yet this can introduce redundant details and impede the model’s ability to learn the features of shrimp larvae. We marked the heads of the shrimp larvae as shown in Figure 15b. This annotation method concentrates on the feature of the shrimp larvae, making it easier to distinguish overlapping larvae. The comparison algorithms included the YOLOv5 algorithm (which means using YOLOv5 without regional segmentation), the Mask R-CNN algorithm [17], and popular commercial software (Smart Shrimp Farm V2.1.3 [51]). In addition, the proposed algorithm was performed on an NVIDIA RTX 3080Ti (NVIDIA, Santa Clara, CA, USA). A total of 500 epochs were trained, with the learning rate set to 0.001. The other parameters were the default YOLOv5 parameters. The optimizer was the stochastic gradient descent.

We used images of shrimp larvae of different densities to evaluate the performance of the proposed method. Under the fixed field of view provided (30 cm × 30 cm), we examined four different densities of shrimp larvae: approximately 1000 larvae for density level 1, approximately 2300 larvae for density level 2, approximately 4000 larvae for density level 3, and roughly 5000 larvae for density level 4. We chose 100 × 100 pixels for the segmentation image size in the counting experiments hereafter. The dataset and codes can be requested by email (Hongchao Duan: 2021024069@m.scnu.edu.cn).

### 4.2. Evaluation Criterion

To quantitatively evaluate the performance of the proposed algorithm, we define *A* as the measurement for the counting accuracy of the formula as follows:(3)$$A=\frac{A_{c}-\left|I_{d}-A_{c}\right|}{A_{c}}\times 100\%$$
where $A_{c}$ represents the actual number of shrimp larvae and $I_{d}$ denotes the number of shrimp larvae.

For accurate results, 15 images of shrimp larvae were analyzed at each density level, with the mean count accuracy serving as the statistical result for each level.

### 4.3. Experiment Comparison

To verify the effectiveness of the above proposed method, we collected a large number of shrimp images of different densities (four densities mentioned in Section 4.1) for the validation experiment. Figure 16a–d shows the recognition effect of the proposed method under different densities. It can be seen in Figure 16 that the proposed algorithm had a good recognition effect for different densities, especially in places where shrimp larvae were densely populated.

Figure 17a shows a typical local image of the original high-density shrimp larvae (approximately 5000 shrimp larvae in a 30 cm × 30 cm area), and Figure 17b shows the corresponding identification results. It shows that the proposed method exhibited high recognition accuracy even when the density of the shrimp larvae was high and there were overlaps between larvae. This indicates that the regional segmentation approach adopted in this study makes the receptive field more accurate, which makes the improved YOLOv5 model more accurate in the recognition of shrimp larvae targets.

Among the various comparison algorithms, Smart Shrimp Farm only provided the final count results without the detection images. We tested high-density shrimp larvae images (approximately 5000 shrimp larvae in a 30 cm × 30 cm area) using the remaining algorithms, and the comparison of the test results is shown in Figure 18. For Density Map Regression, we provide the corresponding density map in Figure 18f.

Table 1 presents the statistical results of the average counting accuracy *A* for the comparison algorithm.

The results clearly indicate that, compared with commercial software such as YOLOv5, YOLOv10, Shrimpseed_Net, Density Map Regression, and Mask R-CNN, the proposed algorithm excelled in counting shrimp larvae at various densities, particularly in high-density scenarios. Although the accuracy decreased somewhat as the density increased, the overall recognition accuracy of the proposed method remained above 98%, significantly outperforming the other three cases (just above 80%). Moreover, the accuracy, which exceeded 98%, demonstrates that the proposed algorithm could potentially substitute the manual shrimp counting process on farms.

### 4.4. Ablation Study

To validate the superiority of the improved YOLOv5 model proposed in this article, we compared the identification results in the segmentation images of 100×100 pixels using both the original YOLOv5 model and the improved YOLOv5 model in Section 3.3 (typical results are shown in Figure 19).

As shown in Figure 19, the improved YOLOv5 model demonstrated better recognition performance when the shrimp larvae were densely packed. To quantitatively analyze the differences between the two models, we used the counting algorithm on all images, and the results of the average counting accuracy for the four density levels are summarized in Table 2. For density 4, we collected images of shrimp larvae under three different lighting conditions (bright, normal brightness, and dim) for testing. The test results are shown in Figure 20, and the data are summarized in Table 2.

The statistical results in Table 2 show that the improved YOLOv5 model achieved an approximately 2% higher counting accuracy compared with the original YOLOv5 model, validating the effectiveness of the proposed method. Under different lighting conditions, the counting accuracy of our proposed algorithm exceeded 98%, outperforming the original YOLOv5 algorithm.

The original image of the larvae was divided into many local areas to decrease the size of the receptive field in Section 3.2. To evaluate the impact of segmentation image sizes on the accuracy of shrimp larvae detection, we cropped the original shrimp larvae images into five different sizes: 600×600 pixels, 300×300 pixels, 200×200 pixels, 100×100 pixels, and 50×50 pixels (as shown in Figure 21). Subsequently, we used the improved YOLOv5 model to train and test these cropped images of different sizes separately.

As shown in Figure 21, when the segmentation image size was 600×600 pixels, many shrimp larvae were not identified. As the segmentation size decreased, the number of undetected shrimp larvae also decreased. When the segmentation image size was reduced to 100×100 pixels and 50×50 pixels, all shrimp larvae in the local images were detected. To quantitatively analyze the differences between the various segmentation image sizes, we tested the counting accuracy using images with four different densities of shrimp larvae mentioned in Section 4.1, and the statistical results of the average counting accuracy *A* for each level of density of shrimp larvae are presented in Table 3.

The results indicate that the size of the image segmentation had a significant impact on the accuracy of the counting of shrimp larvae. As the segmentation image size decreased, the accuracy of the shrimp larvae count clearly increased. When the segmentation image size was reduced to 50×50 pixels, the counting accuracy was the best, but we also realize that the cost of computing increased exponentially. Considering the calculation cost and the counting accuracy, we chose 100×100 pixels for the segmentation image size in the counting experiments hereafter.

### 4.5. Discussion

Comparative experiments showed that the proposed region segmentation algorithm improved the accuracy of counting shrimp larvae. In contrast to the standard YOLOv5 model, the enhanced YOLOv5 variant provides more accurate detection of small objects such as shrimp larvae. Validation experiments indicated that the proposed algorithm performed better, with a counting accuracy of 98% even under conditions with high densities and large volumes of shrimp larvae. However, there are still aspects which need to be improved. To begin with, the dataset for this experiment was gathered using an optical platform which we designed. To improve the generalizability of the algorithm, it is essential to include images of shrimp larvae obtained from various environments and a range of imaging devices in the training dataset. In addition, the stitching and deduplication process occasionally overlooks a few shrimp larvae due to stitching errors, suggesting that this algorithm needs further improvement. In conclusion, the proposed algorithm can be integrated into shrimp larvae imaging devices to develop a comprehensive shrimp larvae counting system.

The image segmentation approach and the improved YOLOv5 model proposed in this paper not only solve the problem of shrimp larvae counting but also have significant research potential in various fields, such as dense crowd counting and medical image analysis. Using segmentation algorithms, the proportion of the pixel of the target objects in the input images increased, thereby improving the detection accuracy. However, the algorithm needs to be adjusted according to the specific problem at hand.

## 5. Conclusions

We introduced a method for counting shrimp larvae which uses region segmentation along with an improved YOLOv5 model. By segmenting the regions, we can reduce the receptive field area, thereby enhancing the detection accuracy of shrimp larvae using the improved YOLOv5 model. Furthermore, the deduplication model can tackle the problem of repeated counts. Based on this, the experimental results show that the number of larvae was about 5000, and the counting accuracy of our algorithm remained above 98%, which would be suitable for replacing the work of manually counting shrimp larvae.

## Figures and Tables

**Figure 1 sensors-24-06328-f001:**
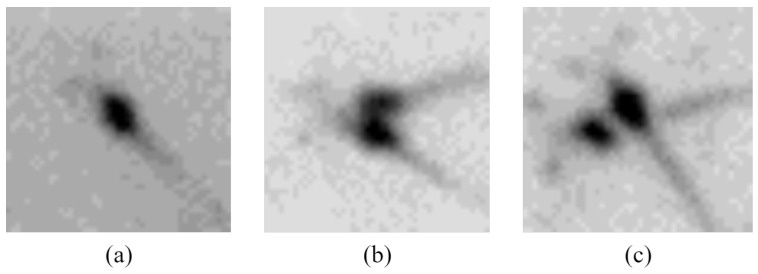
The states of shrimp larvae. (**a**) Isolated shrimp larvae. (**b**) Clumped shrimp larvae. (**c**) Overlapping shrimp larvae.

**Figure 2 sensors-24-06328-f002:**
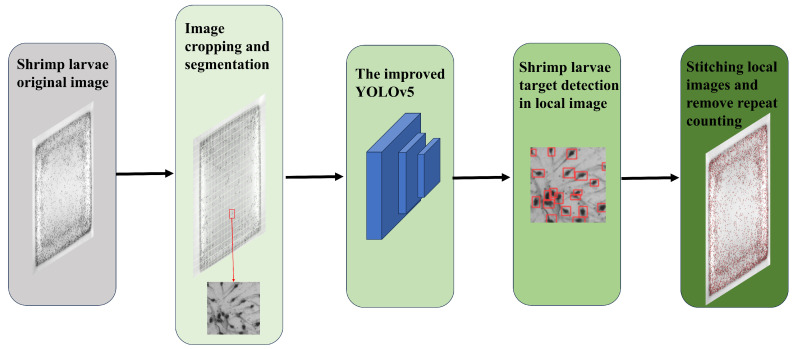
The flow chart of the proposed counting algorithm.

**Figure 3 sensors-24-06328-f003:**
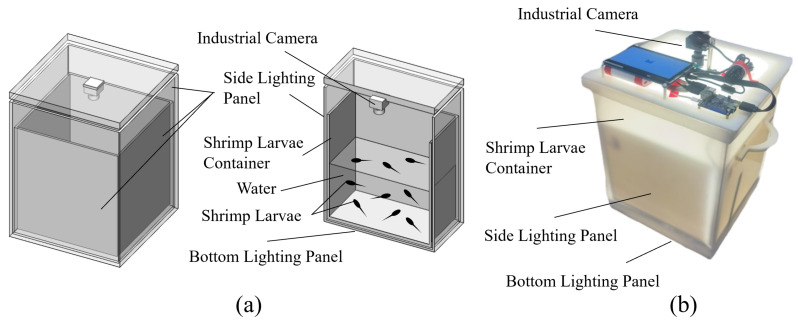
Shrimp larvae acquisition device. (**a**) Schematic diagram. (**b**) Photographs of the actual device.

**Figure 4 sensors-24-06328-f004:**
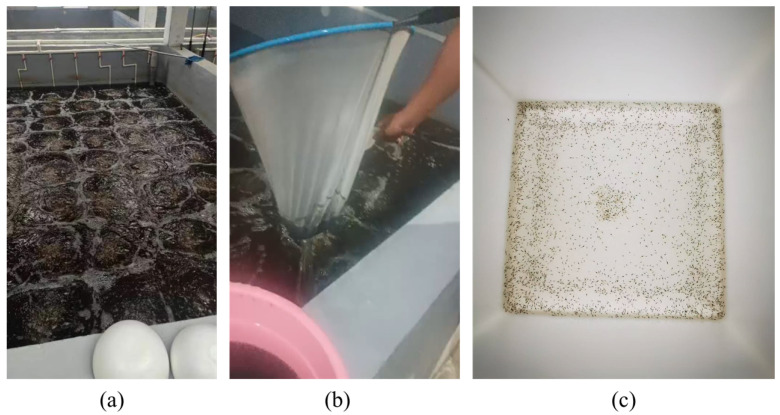
Acquisition of shrimp larvae datasets in a real aquaculture scenario. (**a**) Shrimp culture tank. (**b**) Shrimp larvae collection. (**c**) Acquisition device.

**Figure 5 sensors-24-06328-f005:**
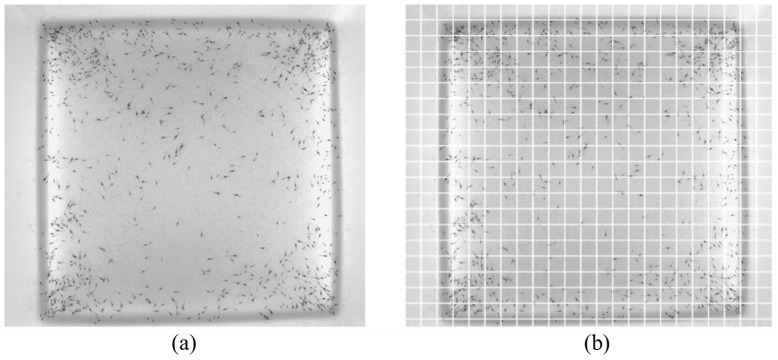
Image segmentation. (**a**) Original shrimp larvae image. (**b**) Segmented image.

**Figure 6 sensors-24-06328-f006:**
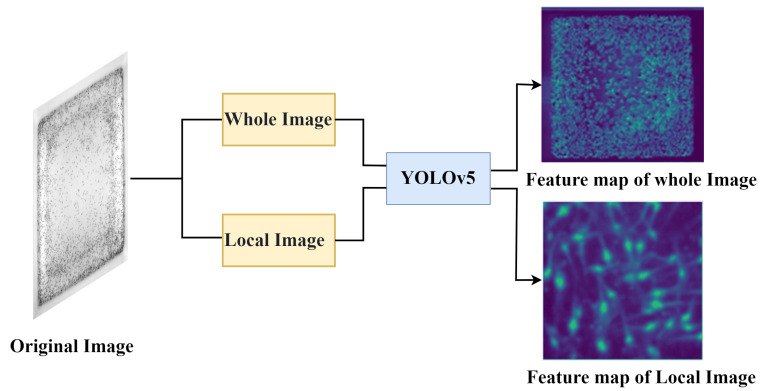
The feature map referring to the shrimp larvae image.

**Figure 7 sensors-24-06328-f007:**
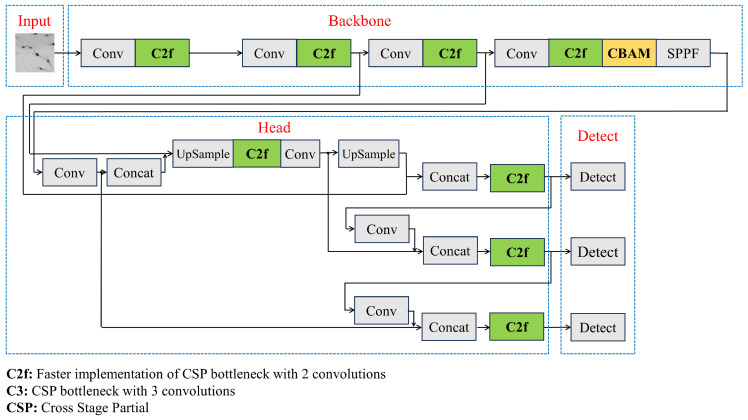
The structure of the improved YOLOv5 model.

**Figure 8 sensors-24-06328-f008:**
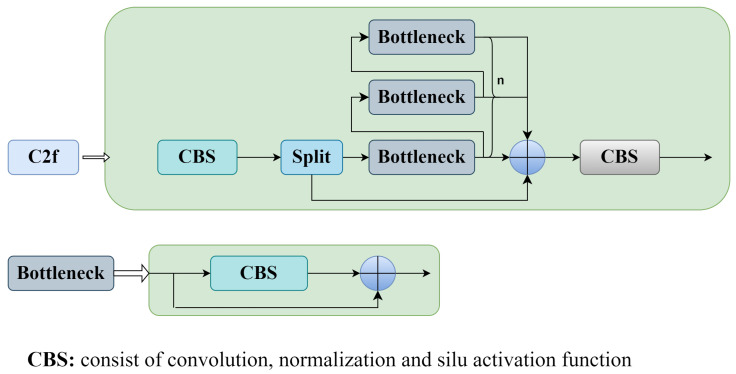
Structure of the C2f network.

**Figure 9 sensors-24-06328-f009:**
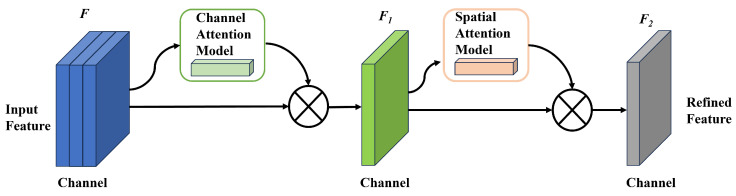
Structure illustration of CBAM.

**Figure 10 sensors-24-06328-f010:**
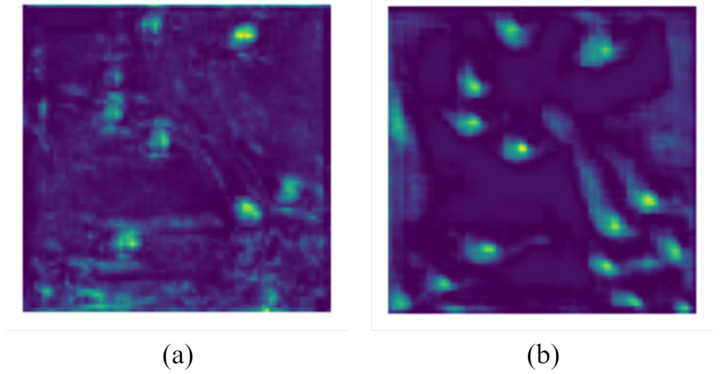
Feature map comparison. (**a**) The original YOLOv5 network. (**b**) The improved YOLOv5 network.

**Figure 11 sensors-24-06328-f011:**
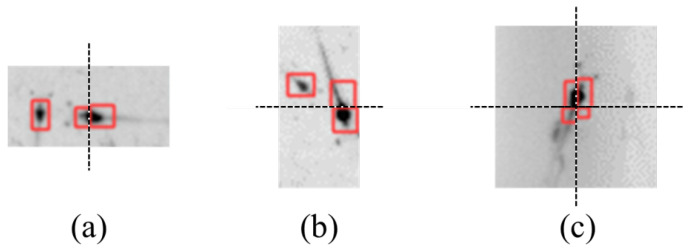
Segmentation of same shrimp larvae in adjacent image blocks (dashed lines indicate the stitching positions). (**a**) Vertical segmentation. (**b**) Horizontal segmentation. (**c**) Both horizontal and vertical segmentation.

**Figure 12 sensors-24-06328-f012:**
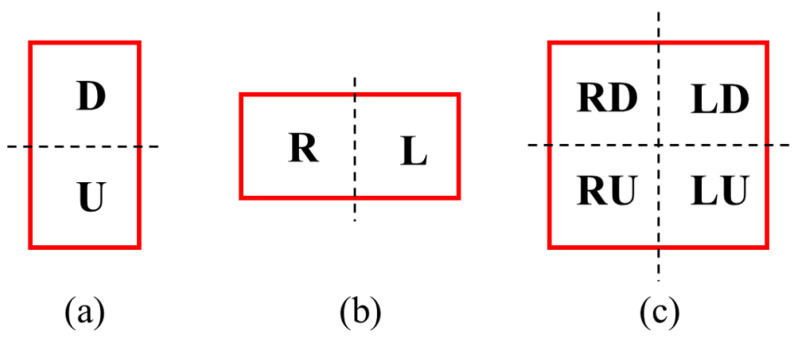
Schematic diagram of shrimp larvae segmentation. (**a**) Vertical segmentation. (**b**) Horizontal segmentation. (**c**) Both horizontal and vertical segmentation.

**Figure 13 sensors-24-06328-f013:**
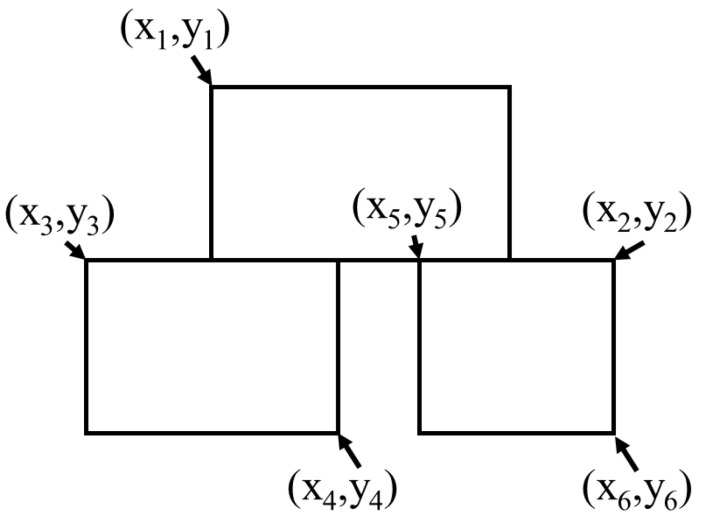
Multiple detection box matching.

**Figure 14 sensors-24-06328-f014:**
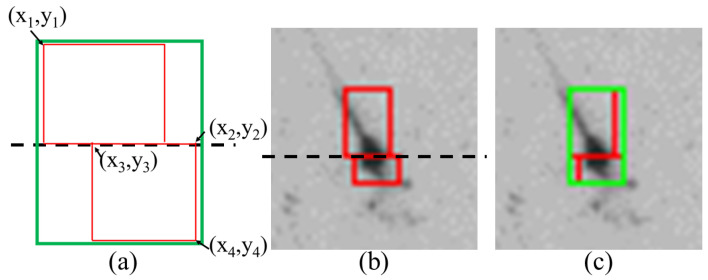
The deduplication area. (**a**) Schematic diagram of how to find the deduplication area (the dashed line indicates region segmentation positions), where the two red rectangles are the detection boxes and the corresponding minimum enclosing rectangle is drawn in green. (**b**) Original detection result with two detection boxes detecting the same larva. (**c**) Minimum enclosing rectangle for the case in (**b**).

**Figure 15 sensors-24-06328-f015:**
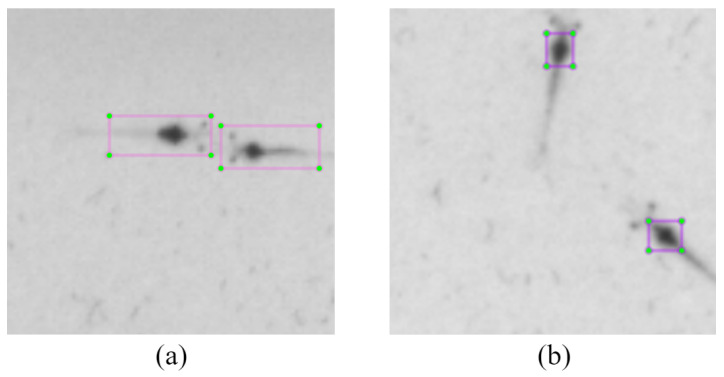
Dataset annotation. (**a**) The traditional labeling method. (**b**) The annotation method we used.

**Figure 16 sensors-24-06328-f016:**
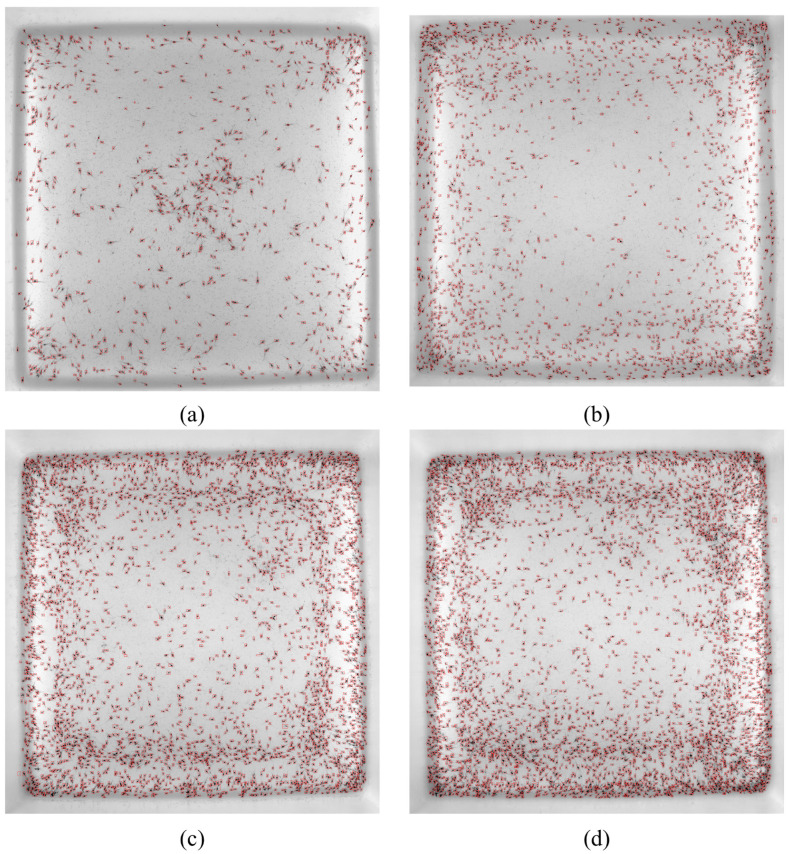
Counting results at different densities. (**a**) Density 1. (**b**) Density 2. (**c**) Density 3. (**d**) Density 4.

**Figure 17 sensors-24-06328-f017:**
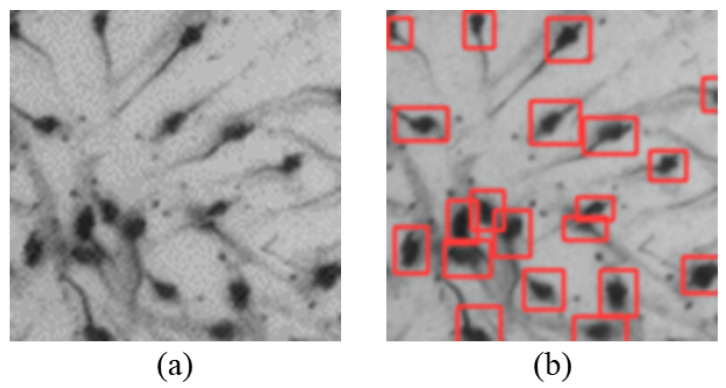
A typical local area of Figure 16. (**a**) Original image. (**b**) Detection results.

**Figure 18 sensors-24-06328-f018:**
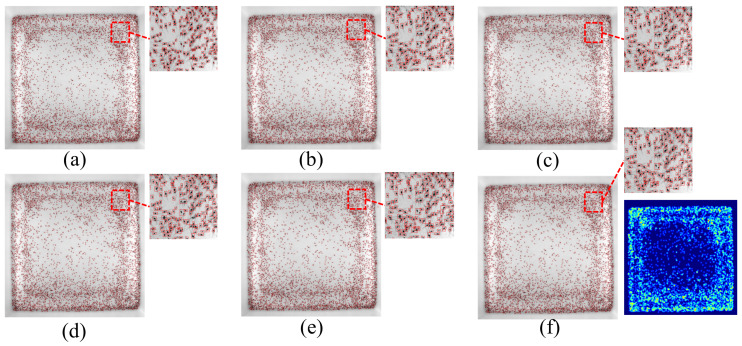
Test result images for different algorithms. (**a**) The proposed method. (**b**) YOLOv5. (**c**) Mask R-CNN. (**d**) YOLOv10. (**e**) Shrimpseed_Net. (**f**) Density map regression.

**Figure 19 sensors-24-06328-f019:**
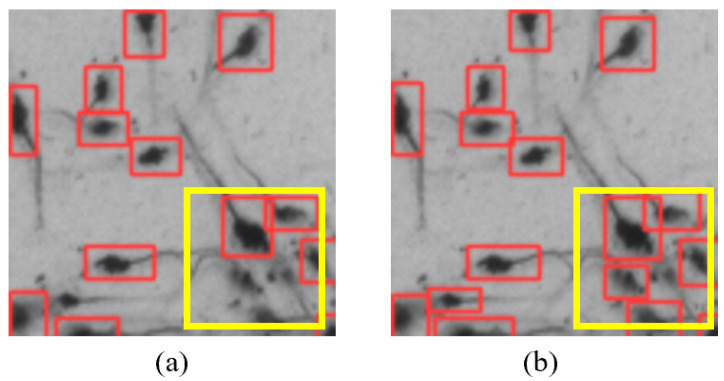
Recognition performance of different algorithm models. (**a**) Original YOLOv5 model. (**b**) Improved YOLOv5 model.

**Figure 20 sensors-24-06328-f020:**
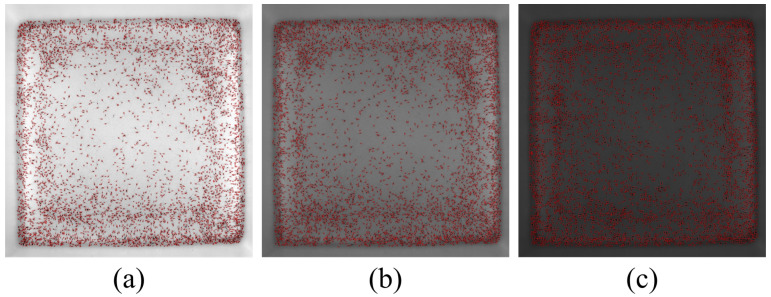
Test results under three different lighting conditions. (**a**) Bright. (**b**) Normal brightness. (**c**) Dim.

**Figure 21 sensors-24-06328-f021:**
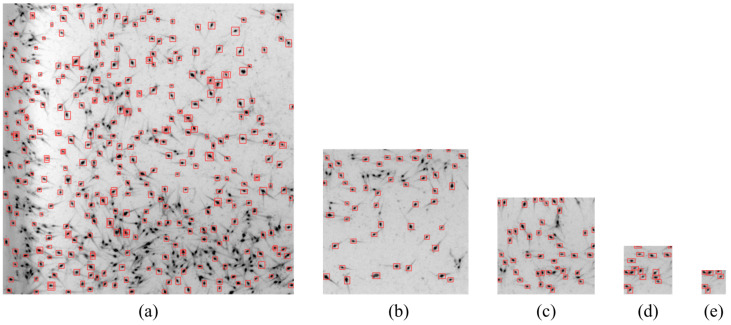
Recognition results for different segmentation image sizes: (**a**) 600×600 pixels, (**b**) 300×3000 pixels, (**c**) 200×200 pixels, (**d**) 100×100 pixels, and (**e**) 50×50 pixels.

**Table 1 sensors-24-06328-t001:** Average counting accuracy *A* for different shrimp larvae densities.

Method	Density 1	Density 2	Density 3	Density 4
**The Proposed Method**	**99.32**%	**99.17**%	**98.53**%	**98.23**%
**Smart Shrimp Farm [51]**	91.43%	90.35%	88.25%	82.99%
**YOLOv5 [52]**	87.79%	85.46%	83.66%	81.03%
**Mask R-CNN [17]**	86.63%	84.52%	81.25%	80.32%
**YOLOv10 [44]**	91.41%	90.68%	89.29%	88.75%
**Shrimpseed_Net [53]**	91.65%	90.23%	89.47%	87.62%
**Density Map Regression [23]**	92.33%	91.58%	90.75%	88.61%

**Table 2 sensors-24-06328-t002:** Average counting accuracy *A* for the two YOLOv5 models.

Density	The Original YOLOv5	The Improved YOLOv5
Density 1	98.13%	**99.32**%
Density 2	97.85%	**99.17**%
Density 3	96.81%	**98.53**%
Density 4 (Bright)	96.26%	**98.23**%
Density 4 (Normal Brightness)	94.66%	**98.21**%
Density 4 (Dim)	93.52%	**98.17**%

**Table 3 sensors-24-06328-t003:** Average counting accuracy *A* for different segmentation sizes.

Density	600 × 600 Pixels	300 × 300 Pixels	200 × 200 Pixels	100 × 100 Pixels	50 × 50 Pixels
Density 1	91.69%	94.27%	97.79%	**99.32**%	99.34%
Density 2	90.83%	93.85%	97.24%	**99.17**%	99.20%
Density 3	88.72%	93.31%	96.38%	**98.53**%	98.87%
Density 4	88.15%	92.56%	95.93%	**98.23**%	98.21%

## Data Availability

Data are available from the authors on request.

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
