# Peer review of "Shrimp Larvae Counting Based on Improved YOLOv5 Model with Regional Segmentation"

_sensors, 2024, doi:10.3390/s24196328_

Round 1

Reviewer 1 Report

Comments and Suggestions for Authors

In this paper the authors proposed an enhanced YOLOv5-based algorithm for counting densely packed shrimp larvae, combined with regional segmentation to improve detection accuracy. The improved YOLOv5 model integrates C2f and attention modules for better recognition of small shrimp. Furthermore, a strategy for stitching and deduplication is also proposed to solve the double counting problem.  The numerical experiments show the superiority of the proposed algorithm, when compared with other shrimp counting techniques, in terms of accuracy.

The paper is interesting and the results are good. 

Minor corrections:

1.      authors should give the meaning of acronyms used in the paper

2. lines 85, 223, 287: It is missing a space between words (ex: “...methods.Clumped ….”

3.      line 97 : should be correct to “….detection [28]. ‘’

4.      in Figure 4 font size of (a) and (b) should be reduced.

5.      In Figure 7, font size of the CBS meaning should be reduced

6.    Figures 12 and 13 should be improved. It is not clear what the coordinates (X_i,Y_i) of each box are and, consequently, to understanding the explanation of the method given in text

7.      In section 4: subsubsections should be subsections (reviewer suggests removing 4.1 since it contains no text)  

8.      In 4.1.1: since the learning rate is given, the authors should also provide the optimizer used in the experiments.

9.      Figure 15: the legend should be corrected ( … results; 1000;  mean instead about?)

Reviewer 2 Report

Comments and Suggestions for Authors

This manuscript introduced an enhanced YOLOv5 model through a regional segmentation approach for counting densely packed shrimp larvae. The authors build the C2f and convolutional block attention modules, and employ a regional segmentation technique. The experimental results confirm the effectiveness of this method.

However, several issues:

1. Although the abstract mentions that “Counting shrimp larvae is an essential part of shrimp farming.”, Could the algorithm be tested in real case conditions? Are the results presented representative of real case conditions? Can the algorithm be applied in practice?

2. As I know solving small object counting problem using density map estimation is more effective, why did you choose object detection method. Besides, what is the advantage of object detection method over density map estimation method in small object? Please add the counting method of density map estimation as a comparison method to verify the performance of the proposed method.

3. Why not YOLOv10 instead of YOLOv5? What is the performance with YOLOv10?

4. Please validate the proposed method by comparing it with several state-of-the-art counting methods.

5. In a real shrimp culture scenario, shrimp larvae live in dark lighting and turbid conditions. How about the performance of the proposed method in the actual breeding scene?

6. The discussion on the proposed potential for applications in other domains, such as people counting and medical image analysis, is insightful. Further elaboration on these applications, their advantages and potential limitations would be beneficial.

7. I would encourage you sharing your algorithm and datasets in a public repository so that your work could be repeated by others and further contribute to the scientific community.

Reviewer 3 Report

Comments and Suggestions for Authors

The paper explores a shrimp larvae counting method based on an improved YOLOv5 model and region segmentation technology, but there are the following issues:

1.       The YOLO series has developed to v10 version. Why did the author still choose v5 version? It is suggested to explain this in the article;

2.       Suggest presenting the contributions of this article one by one at the end of the Introduction;

3.       Section 3.2 introduces the feature extraction of segmented images, but does not provide detailed information on how to fuse the features of whole and local. If there are shrimp larvae at the edges of the segmented images, how should they be processed?

4.       The experiment only demonstrated the effectiveness of the proposed method and did not compare it with the latest methods. It is recommended to add corresponding experiments;

5.       The collected image scenes are relatively single, and experimental comparisons should be conducted under different lighting conditions and different viewing angles;

6.       The conclusion lacks quantitative indicators, it is recommended to add them.

Comments on the Quality of English Language

Partial statements can be more concise.
